# Two-Stage Bio-Hydrogen and Polyhydroxyalkanoate Production: Upcycling of Spent Coffee Grounds

**DOI:** 10.3390/polym15030681

**Published:** 2023-01-29

**Authors:** Beom-Jung Kang, Jong-Min Jeon, Shashi Kant Bhatia, Do-Hyung Kim, Yung-Hun Yang, Sangwon Jung, Jeong-Jun Yoon

**Affiliations:** 1Green & Sustainable Materials R&D Department, Korea Institute of Industrial Technology (KITECH), Chunan-si 31056, Republic of Korea; 2Department of Biological Engineering, Konkuk University, Seoul 27478, Republic of Korea; 3Sustainable Technology and Wellness R&D Group, Korea Institute of Industrial Technology (KITECH), Jeju-si 63243, Republic of Korea; 4Department of Bio and Fermentation Convergence Technology, Kookmin University, Seoul 02707, Republic of Korea

**Keywords:** bio-hydrogen, polyhydroxyalkanoate, mcl-PHA, *Pseudomonas resinovorans*

## Abstract

Coffee waste is an abundant biomass that can be converted into high value chemical products, and is used in various renewable biological processes. In this study, oil was extracted from spent coffee grounds (SCGs) and used for polyhydroxyalkanoate (PHA) production through *Pseudomonas resinovorans*. The oil–extracted SCGs (OESCGs) were hydrolyzed and used for biohydrogen production through *Clostridium butyricum* DSM10702. The oil extraction yield through *n*–hexane was 14.4%, which accounted for 97% of the oil present in the SCGs. OESCG hydrolysate (OESCGH) had a sugar concentration of 32.26 g/L, which was 15.4% higher than that of the SCG hydrolysate (SCGH) (27.96 g/L). Hydrogen production using these substrates was 181.19 mL and 136.58 mL in OESCGH and SCGH media, respectively. The consumed sugar concentration was 6.77 g/L in OESCGH and 5.09 g/L in SCGH media. VFA production with OESCGH (3.58 g/L) increased by 40.9% compared with SCGH (2.54 g/L). In addition, in a fed–batch culture using the extracted oil, cell dry weight was 5.4 g/L, PHA was 1.6 g/L, and PHA contents were 29.5% at 24 h.

## 1. Introduction

Coffee is the world’s most-consumed drink, and about 167.17 million metric tons of coffee were produced in 2021 [1]. This is three times the world’s average coffee consumption, and about 190,000 metric tons of green beans were imported in 2021 [2]. The problem with coffee consumption is that spent coffee grounds (SCGs) are produced in large quantities. After extracting the drink from the coffee bean, a large amount of SCGs remain that are known to have a dry weight of about 65% of the initial cherries [3]. Based on import figures, it is estimated that about 150,000 metric tons of SCGs were produced in Korea in 2021. Most of the SCGs are dumped as garbage and buried or incinerated [4]. Since coffee is not produced in Korea, importing coffee is tantamount to importing garbage and means that coffee imports lead to 150,000 metric tons of waste every year. From an environmental engineering perspective, SCGs are organic waste resources, of which the main components are carbohydrates, proteins, and fats. Therefore, it is necessary to devise a plan to utilize SCGs in various ways.

Organic waste and biomass resources (rice straw, marine algae, food waste, etc.) usually have a large amount of carbohydrates that can be used by microorganisms, so they are used in the form of monosaccharides or oligomers through hydrolysis [5,6,7]. This can be applied equally to SCGs, and there are several studies on using SCGs as a resource. Studies have been conducted to utilize SCGs that have undergone hydrolysis to produce butanol, ethanol, methane, and polyhydroxyalkanoate (PHA) [8,9,10]. However, the SCGs were not fully utilized by hydrolysis only. As SCGs are composed of carbohydrates (about 60%), proteins (about 15%), and fats (about 15%) [4,11], it is possible to devise a plan to separate and utilize the oil among the ingredients of SCGs. In previous studies, SCG oil was extracted and utilization measures such as PHA production were devised [9]. However, the studies only used SCG oil, and they did not fully use SCGs. Therefore, the extracted oil and the hydrolysates of the oil–extracted SCGs may be utilized to increase the value of the SCGs.

Recently, interest in hydrogen energy and bioplastics is increasing because of global issues such as environmental pollution and fossil fuel depletion [12,13]. Hydrogen is one of the promising eco-friendly resources, and has high energy yield of 122 kJ/g which is 2.75 times greater than that of hydrocarbon fuel [14]. It can be generated by fossil fuel, electrolysis of water or biological processes—among them, bio-hydrogen, which can be produced through utilization of biomass hydrolysates by microbes [15]. Therefore, various biomass hydrolysates such as corn stover, marine algae, oil palm empty fruit bunches, rice straw and sugarcane bagasse were used to produce bio-hydrogen via dark fermentation, resulting in 0.76 to 2.24 hydrogen yield (mol/mol) [14,15,16,17]. As in the above-mentioned studies of bio-hydrogen production from various biomass hydrolysates, oil-extracted SCGs could be considered an alternative carbon source from which to produce bio-hydrogen.

Bioplastics are eco–friendly plastics that can be produced or decomposed by microorganisms [18]. Similar to fossil-fuel based plastics, they offer a variety of physical properties. PHA, a representative bioplastic, is a polymer in which 3–hydroxyalkanoate produced by microorganisms for storing carbon sources is ester–bonded under unfavorable conditions for growth [19]. PHA is classified depending on the number of carbon atoms in the monomers as short chain length (scl, C3–5), medium chain length (mcl, C6–14), and long chain length (lcl, C15–18) PHA; additionally, the type and physical properties of PHA vary depending on the metabolism or substrate provided by the bacteria [20]. Many studies have been conducted on scl–PHA and mcl–PHA because of their physical properties and production yield advantages; scl–PHA also has solid properties in the form of polymers such as 3-hydroxybutyrate (3HB) and 3-hydroxyvalerate (3HV) [21], while mcl–PHA has elastic and adhesive properties [18]. In order for bioplastics to have various physical properties and forms as with fossil fuel–based plastics, it is necessary to promote research on mcl–PHA, which shows relatively low yield. Mcl–PHA can be produced through various bacteria and with different carbon sources. Among such bacterial strains, *Pseudomonas* sp. is a representative strain known to produce mcl–PHA [22]. Several studies have shown that *Pseudomonas* sp. is able to use fatty acids such as olive oil and waste cooking oil, thereby enhancing the value of various organic waste resources [23].

Therefore, this study aims to (i) investigate SCG oil extraction and yield, (ii) hydrolyze oil–extracted SCGs and use them as hydrogen production substrates, (iii) investigate PHA production possibilities using extracted SCG oil, and (iv) investigate the possibility of mass culture using a 10 L fermenter.

## 2. Materials and Methods

### 2.1. Materials

All reagents and chemicals were procured at the highest purity. M9 minimal salts (5×), 3-Hydroxyoctanoic acid, 3-Hydroxydecanoic acid, and 3-Hydroxydodecanoic acid were purchased from Sigma Aldrich (St. Louis, MO, USA). Chloroform, H_2_SO_4_, *n*-hexane, HCl, NH_4_Cl, NaOH, methanol, NaCl, sodium acetate, sodium bicarbonate, palmitic acid, oleic acid, stearic acid, linoleic acid, l–cysteine hydrochloride and antifoam were purchased from Daejung Chemical & Metals (Siheung-si, Republic of Korea). LB media, yeast extract, peptone, beef extract, soybean extract and urea were purchased from BD-DIFCO (Detroit, MI, USA).

### 2.2. SCGs Oil Extraction

SCGs were obtained from a local cafe in Cheonan-si, South Korea. To dry the SCGs, they were spread widely and sun–dried for 8 h and then dried in a drying oven at 105 °C for 24 h. Oil was extracted from the dried SCGs using a Soxhlet extractor with 200 mL of *n*–hexane as a solvent, and 25 g of SCGs was put into a weighed cellulose filter for oil extraction for 1.5 h. The heating mantle was kept at 70 °C. After oil extraction was completed, the filter containing SCGs was dried in a fume hood for 24 h to evaporate the *n*–hexane and then weighed. The oil mixed with the *n*–hexane was purified by evaporation at 50 °C and 50 rpm for 30 min using a rotary evaporator. The concentrated SCGs oil was weighed to confirm the amount of the *n*–hexane residue and compared with the SCGs weight change. Mass extraction was carried out for future experiments with the products sterilized at 121 °C for 15 min in an autoclave and then refrigerated.
Weight of SCGs extract = (filter weight + SCGs weight) − (dried filter weight + SCGs weight after oil extraction)(1)
The amount of *n*–hexane remaining in SCGs oil (*w*/*w*%) = (SCGs oil weight − SCGs weight variation)/SCGs oil weight × 100(2)
SCGs oil extraction yield (%) = SCGs weight variation/SCGs initial weight × 100(3)

Gas chromatography–mass spectrometry (GC/MS; Claus 500, Perkin Elmer, Waltham, MA, USA) analysis was performed to analyze the main components of the extracted SCGs oil, and a gas chromatograph (GC; 6890N, Agilent Technologies, Santa Clara, CA, USA) was used for quantitative analysis.

### 2.3. Oil–Extracted SCGs Hydrolysis and Hydrogen Production

The oil–extracted SCGs (OESCGs) were used as a substrate for hydrogen production. OESCGs were dried in the fume hood for 12 h to remove residual *n*–hexane. Then, hydrolysis was performed with a 10% solid/liquid (S/L) ratio and 1% H_2_SO_4_ at 130 °C for 1 h. OESCG hydrolysate (OESCGH) was centrifuged at 3500 rpm for 30 min, and the supernatant was recovered. The composition of the OESCGH was analyzed through high–performance liquid chromatography (HPLC; 1200 series, Agilent Technologies, Santa Clara, CA, USA). The OESCGH was mass–produced and then sterilized in an autoclave for 15 min at 121 °C before being refrigerated for use in subsequent experiments.

An experiment was conducted to compare the hydrogen productivity of general SCG hydrolysate (SCGH) and OESCGH. The culture medium was defined–reinforced clostridium medium (RCM). Here, SCGH or OESCGH was added and diluted to a total sugar concentration of 10 g/L. *Clostridium butyricum* DSM10702 was used for hydrogen production and was inoculated in the anaerobic chamber under 99.9% of N_2_. The culture conditions were pH 5.5 (adjusted by HCl), 37 °C, and 150 rpm. Gas production, hydrogen production, and metabolite change were measured for 32 h. Defined RCM was composed as follows: 3 g/L yeast extract, 10 g/L peptone, 10 g/L beef extract, 0.5 g/L L–cysteine hydrochloride, 3 g/L sodium acetate, 5 g/L sodium chloride, 5 g/L sodium bicarbonate, and 100 μL/L antifoam.

### 2.4. PHA Production Using SCGs Oil

A PHA production test was conducted to confirm the suitability of the SCGs oil. *Pseudomonas resinovorans*, which is known to produce mcl–PHA using oil as a substrate, was used as the inoculum. A stock stored in a deep freezer was pre–cultured in 5 mL of lysogeny broth (LB) media. Pre–cultured bacteria were inoculated into the main culture medium after 24 h. The medium was M9 salt medium, and 2% oil was added as a substrate. After that, for C/N ratio optimization, various N sources were added at 0.1% and 0.5%, and productivity was confirmed. A fed–batch culture was performed using ammonium chloride as an N source and SCGs oil as a substrate. The reactor was a 10 L fermenter (BIOCNS Co., Ltd., Deajeon, Republic of Korea), with a working volume of 3 L, in duplicate, and inoculation volume of 10%. The medium was M9 salt medium, initial SCGs oil concentration was 2%, and SCGs oil was fed at 5% at 4 h and at 18 h. An amount of 1 M HCl and 3 M NaOH were used to adjust the pH to 6.8 ± 0.2, and aeration was maintained at 10 L/3 L/1 min to adjust to 30% dissolved oxygen (DO). The reactor was operated at 600 rpm and 30 °C for 96 h, and samples were taken to analyze cell dry weight (CDW) and PHA contents. The culture medium was centrifuged at 4 °C and 3500 rpm, for 1 h. The supernatant was discarded, and the cell pellets were freeze–dried after washing. Acetone was added to the freeze–dried cells to dissolve PHA, and methanol was treated at 1:10 to recover PHA precipitated at the bottom.

### 2.5. Analysis

GC/MS was used for qualitative analysis using the method described in previous studies [24]. After obtaining fatty acid methyl ester (FAME) from the SCGs oil, the analysis was conducted. Peaks were identified by the mass spectrometric fragmentation data and confirmed by comparison to spectral data that was available from the online libraries of Wiley (http://www.palisade.com, accessed on 12 December 2022) and NIST (http://www.nist.gov, accessed on 12 December 2022).

For quantitative analysis of the SCGs oil, fatty acid (for standard curve), and PHA, each sample was analyzed using the GC. Samples of 10 mg of SCGs oil or fatty acid were contained in a 15 mL glass round bottom tube for pretreatment for FAME analysis.

For quantitative analysis of PHA, 1 mL of culture medium was centrifuged at 13,000 rpm for 5 min. Afterwards, the cell pellet was washed twice with deionized water (DW), and the resuspended cells were placed in a 15 mL glass round bottom tube that was sealed with teflon, frozen in a freezer, and freeze–dried. Afterwards, the CDW was measured, and the dried cells were treated for FAME analysis. For FAME, 1 mL of chloroform and 1 mL of sulfuric acid–methanol solution (15:85) were added and the sample was reacted at 100 °C for 2 h on a heating block. The sample was then cooled at room temperature for 10 min and then chilled on ice for 10 min. After adding 1 mL of iced cold water, the sample was shaken and the organic phase below was analyzed. Of the prepared sample, 1 µL was injected through an auto–sampler and analyzed with a flame ionization detector (FID).

The analysis conditions were as follows: a 30 m × 0.25 mm DB–FFAP capillary column, hydrogen 40 mL/min as the carrier gas, air zero flow 450 mL/min and high purity nitrogen gas 45 mL/min as the makeup flow, inlet temperature 250 °C, detector temperature 250 °C, and the oven held at 90 °C for 5 min, heated to 220 °C with a heating rate of 20 °C/min, and then held at 220 °C for 7 min.

For analysis of bio-hydrogen productivity, the gas produced in the serum bottle was collected through a 1 mL syringe (Hamilton Company, NV, USA) and injected into the GC. Hydrogen measurement was performed through a thermal conductivity detector (TCD) with a 10 m × 0.3 mm CP–Molsieve 5A capillary column. The analysis conditions were as follows: high purity N_2_ gas (99.999%) was used as the carrier gas, the inlet temperature was 100 °C, the detector temperature was 250 °C, and the oven was held at 80 °C.

HPLC was used to measure metabolic product changes in the bio-hydrogen production. A 1 mL sample was centrifuged for 5 min at 4 °C and 13,000 rpm. Then, the supernatant was filtered through a PTFE membrane filter with 0.45 μm pore size. The HPLC analysis conditions were as follows: 5 mM H_2_SO_4_ as a mobile phase, 60 °C for column temperature, 300 mm × 7.8 mm Aminex HPX–87H ion excursion column, 55 °C for detector temperature, refractive index detector, 25 μL of injection volume, 0.6 mL/min of flow rate, and 55 min of running time.

The measured hydrogen and biogas were converted to values at standard pressure and temperature before being applied to the modified Gompertz model below [25]:(4)$$H=P\times \exp\left\{-exp\left[\left(\left(Rm\times e\right)\right)/P\times \left(\lambda -t\right)+1\right]\right\}$$

H is the estimated hydrogen production (mL), *P* is the hydrogen production potential (mL), *Rm* is the maximum hydrogen production rate (mL/h), λ is the lag phase (h), t is the time (h), and *e* is 2.7182 [25].

## 3. Result and Discussion

### 3.1. Extracted Oil from SCGs

In general, oil extraction from oilseed includes press extraction and organic solvent extraction. The organic solvent extraction method was selected because SCGs are not suitable for the press extraction method as they are finely ground. Solvents used for oil extraction include *n*–hexane, ethanol, methanol, pentane, acetone, and isopropanol [26,27]. According to Al–Hamamre et al. and Pichai et al., various solvents were used to extract SCGs oil, and relatively high yields were found when *n*–hexane was used [26,27,28]. According to Efthymiopoulos et al., the amount of SCGs and *n*–hexane and extraction time used were 25 g, 200 mL, and 1.5 h, respectively [29].

The total weight of the cellulose filter and SCGs was 30.05 g (Table 1). The total weight of the dried cellulose filter and SCGs after oil extraction was 26.43 g, and the SCGs weight variation was 3.62 g. The mixture of *n*–hexane and SCGs oil was fractionally distilled, and the concentrated SCGs oil was 4.03 g. There was a difference of 0.41 g between the concentrated SCGs oil and the SCGs weight variation. It was found that about 10.4% of *n*–hexane in SCGs oil remained even after fractional distillation (data not shown). Therefore, it is important to select strains that can operate efficiently considering that *n*–hexane is known to be toxic to a relatively large number of bacterial species [30].

The SCGs oil extraction yield was 14.4%, and the fat content in the SCGs was 14.9% based on the dry weight (Table 2). Based on these figures, the oil extraction rate was 97.0%. The results of analyzing the components of SCGs oil through GC/MS and GC showed that the SCGs consisted of 34.1% palmitic acid, 16.8% stearic acid, 10.3% oleic acid, and 38.8% linoleic acid (Appendix A). Other studies found that oil accounted for 10–15% of SCGs [29,31]. One of those studies showed that SCGs oil consists of palmitic acid, stearic acid, oleic acid, and linoleic acid [29], which are the same components as for the SCGs used in the present experiment. There was a difference in oil content and composition ratio, and this is believed to be because of different coffee production areas. A large amount of coffee oil was extracted and then sterilized and refrigerated for use in the next experiment. According to previous studies conducted by Al-Hamamreet et al. and others, 14.7% [26] and 15.28% [27] were extracted for 30 min using the Soxhlet extractor. Similar to this study, it was shown that the entirety of the oil in SCGs could be recovered. In addition, although there was a difference in content of fatty acids in SCGs oil at various studies, most of them were palmitic acid, linoleic acid, oleic acid, and stearic acid, showing that they were similar to the results of this study [26].

### 3.2. Hydrolysis of Oil Extracted-SCGs

For hydrogen production experiments, hydrolysis was performed using OESCGs. The OESCGs were hydrolyzed under conditions derived from previous experiments (S/L ratio 10% (*w*/*w*), H_2_SO_4_ 1.0% (*w*/*w*), 130 °C, and 1 h). The hydrolysis results of general SCGs and OESCGs are shown in Figure 1, and the total sugar concentration of the OESCGH was 32.26 g/L (Figure 1A). This is about 15.4% higher than that for the SCGH (27.96 g/L).

In addition, this was similar to the amount when the S/L ratio was 20% (*w*/*w*) and H_2_SO_4_ 1.0% (*w*/*w*) (35.93 g/L), which is thought to be because of an increase in carbohydrate content after oil is extracted. The carbohydrate content of dried SCGs was 69.2% (Table 2), while that of OESCGs was 80.8% (=69.2/[100 − 14.4] × 100).

Comparing the sugar concentration with other studies, Obruca et al. reported that the total sugar concentration was 50.1 g/L under the conditions of 1% H_2_SO_4_, 121 °C, 90 min, and 15% of the S/L ratio [9]. Hudeckova, Helena, et al. showed 23.86 g/L under the conditions of 2.7% H_2_SO_4_, 121 °C, 15 min, and 10% S/L ratio [32]. It was confirmed that the sugar recovery rate varies depending on H_2_SO_4_ concentration, reaction time, and S/L ratio. It was verified that our experimental conditions recovered a relatively high concentration of sugar even that H_2_SO_4_ and S/L ratios were less than the previous experiments.

Furthermore, the monosaccharide recovery rate compared with the carbohydrate content of SCGs (69.2%) was 40.4%, and that of the OESCGs (80.8%) was 39.9%. It was also found that the ratio (3.1:6.9) of galactose and mannose, which are the main sugars of the SCGH, remained the same as that of the general SCGs hydrolysis (3.1:6.9). These results show that hydrolysates of the same composition with higher sugar concentrations can be obtained through oil extraction.

The concentration of short chain fatty acids (SCFAs) tended to be increased, similar to the sugar concentration, with the SCFA concentration of the OESCGH at 2.61 g/L, higher than that of the SCGH (1.86 g/L), whereas the concentrations of the furan derivatives were 0.27 g/L and 0.28 g/L, respectively, with no increasing tendency (Figure 1B,C). This result is similar to that of the previous hydrolysis for each S/L ratio where SCFAs tended to increase as the S/L ratio increased.

It is known that the concentration of furan derivatives that inhibits the growth of microorganisms is low, and it has been shown that bacterial growth is inhibited above 0.5 g/L 5–hydroxyl methyl furfural or above 2.0 g/L furfural [33,34]. In the SCGs hydrolysate, furan derivatives were produced at a concentration that did not cause inhibition of microbial growth, so the post–treatment process could be omitted. The above inhibitors generated during biomass hydrolysis are removed through physico–chemical processes such as adsorption using powdered activated carbon. This leads to an increase in process costs. Such a post–treatment process is not required for the SCGH; therefore, it provides process advantages and may be used as an ideal substrate.

### 3.3. Bio–Hydrogen Production Using Oil–Extracted SCGs Hydrolysate

Hydrogen production experiments were conducted using the SCGs hydrolysates prepared above, and hydrogen productivity determined for the SCGs hydrolysates from general SCGs and OESCGs (Figure 2). When using the OESCGH, the lag time was shorter and the bio-hydrogen production was higher than when using the SCGH (Figure 2A–C). Lag time was 4.8 h and 6.1 h, respectively, and hydrogen production was 181.19 mL and 136.58 mL, respectively, in OESCGH and SCGH media. However, there was no significant difference in hydrogen production yield.

There was a large difference in VFA production and sugar consumption between SCGH and OESCGH (Figure 2D). In OESCGH media, the consumed sugar concentration was 6.77 g/L, 33.0% higher than in SCGH media (5.09 g/L). VFA production increased with OESCGH (3.58 g/L) by 40.9% compared with SCGH (2.54 g/L). Although OESCGH media showed a fast lag time, high production of hydrogen and VFA, and high sugar consumption, it seems that the yield was similar to that of the SCGH media and this was because of the input amount of the hydrolysate.

When producing RCM–based media, each hydrolysate was added so that the total sugar concentration was 10 g/L. Accordingly, relatively less was added for the volume of OESCGH because OESCGH had a high sugar concentration. Although sugar concentrations were the same in each medium, there were differences in the amounts of SCFA, furan derivatives, and H_2_SO_4_. One of these factors is believed to have caused the difference in productivity. When SCGH and OESCGH were put in the RCM media in the same volume, there was no significant difference in hydrogen production, sugar consumption, or lag time (Figure 3). Accordingly, the difference in productivity seems to be because of the concentration of sulfate ion from the H_2_SO_4_ and buffer capacity. In summary, OESCGH has a high sugar concentration, which can induce cost reductions and play a good role in increasing productivity. In addition, it shows a high yield compared to various biomass used in the production of bio–hydrogen, although there may be differences depending on the fermentation conditions (Table 3). It is estimated due to the different composition of hydrolystates, because most lignocellulose biomass has glucose, xylose, and arabinose as the main sugar components, but SCGs hydrolysates are mainly composed of mannose and galactose.

### 3.4. Nitrogen Sources and C/N Ratio Balance Comparison

Generally, the PHA accumulation pathway is triggered when the carbon source is sufficient, but nitrogen sources are limited [35]. Considering that SCGs oil exists as micelles during fermentation and is utilized by the β-oxidation pathway, selection of proper nitrogen sources is important. Thus, six different nitrogen sources (ammonium chloride, yeast extract, beef extract, peptone, soybean extract, and urea) were evaluated at concentrations of 0.1% and 0.5%, along with 2% of SCGs oil in M9 minimal media. Ammonium chloride, peptone, and urea showed high cell growth, in the range of 2.6~3.1 g/L of CDW, compared with the other nitrogen sources; most of the nitrogen sources showed similar PHA production in the range of 0.42~0.51 g/L at 0.1% concentration, except yeast extract (Figure 4A). While ammonium chloride and beef extract showed higher cell growth at 3.0 g/L and 3.4 g/L, respectively, compared with the other nitrogen sources, at 0.5% concentration beef extract provided slightly better production of PHA at 0.6 g/L compared with 0.5 g/L of PHA produced with ammonium chloride (Figure 4B). Yeast extract showed a negative effect on PHA production, while it was confirmed that *P. resinovorans* produced PHA with ammonium chloride even without a complex media component.

To investigate the C/N ratio balance when the SCGs oil is used as a carbon source, ammonium chloride was used with concentrations in the range of 0.01 to 1%. Cell growth and PHA production increased up to 0.5% ammonium chloride, reaching 3.0 g/L and 0.5 g/L, respectively (Figure 5). However, with 1% ammonium chloride, cell growth and PHA production were both reduced dramatically. It may be that excessive nitrogen exposure triggered nitrogen regulatory proteins that had a negative effect on cell growth metabolism, or there may be an effect on the β-oxidation pathway in *Pseudomonas* species [36,37].

### 3.5. Fed-Batch Culture for PHA Production

Application of a fed-batch strategy to increase PHA productivity was proved by many other related studies [38,39,40]. Therefore, *P. resinovorans* was cultured with an optimized C/N balance on a flask scale, and cell growth and PHA production monitored (Figure 6A). Cell growth during the fermentation reached 2.7 g/L after 24 h and 7.9 g/L after 72 h. While PHA production tended to increase steadily, increasing to 0.75 g/L after 24 h and 2.4 g/L after 72 h, PHA content was maintained at around 30%. It may be that knock-out of PHA-degrading enzymes such as *phaZ* or optimization of the PHA accumulation pathway is required in order to increase the PHA content further.

The fed-batch culture was performed through pulse feeds on a 10 L scale jar fermenter. In particular, considering the long lag phase of *P. resinovorans* with the flask scale, 5% SCGs oil was added after 4 h to boost cell growth, and the same concentration of SCGs oil was then added at 18 h. From the results, it was confirmed that the lag phase was much shorter than with the flask scale, and CDW reached 4.4 g/L at 12 h and 5.4 g/L at 24 h of the fermentation (Figure 6B). In addition, PHA production increased at the same time as the growth phase, increasing from 0.6 g/L at 12 h to 1.6 g/L at 24 h. The PHA content increased to 29.5% within 24 h and remained constant. It is believed that the reason for the decrease in the lag phase was because of pH and DO adjustment.

Compared with previously-reported results of PHA production using SCGs oil, it found carbon sources to be effective in mcl-PHA production (Table 4). However, produced amount mcl-PHA is relatively low than PHB, it is estimated that development of strain in genetic or metabolic engineering level is required to improve PHA production.

### 3.6. Physical Properties of Produced PHA

Differential Scanning Calorimetry (DSC) was used to investigate the thermal properties of mcl-PHA, including the glass transition temperature (Tg), melting temperature (Tm), crystallization temperature (Tc), and melting enthalpy (ΔHm). The results obtained for the scl-co-mcl-PHA were Tg = −59.79 °C and Tc = 86.8 °C with ΔHm = 38.6 J/g, and Tm = 11.239 °C with ΔHm = 41.2 J/g (Appendix A).

The thermal characteristics of our mcl-PHA differed both in Tm and Tg from those of the previously-reported mcl-PHA from *Pseudomonas* sp. PAMC28620, composed of poly(25.5% 3HO-co-52.1% 3HD-co-5.7% 3HdD-co-16.7% 3HtD) with Tm = 172.8 °C, Tg = 3.99 °C, and Tc = 54.61 °C [44]. The thermal properties also differed from those of the mcl-PHA from *Pseudomonas* sp. MPC6. Composed of poly(89.5% 3HB-co-1.8% 3HHx-co-3.3% 3HO-co-4.4% 3HD-co-1.1% 3HdD) with Tm = 163.5 °C, Tg = 2.3 °C, and Tc = 46.0 °C [45].

As highly crystalline polymers have limited application in industrial and medical fields [46], the mcl-PHA from *P. resinovorans* may have potential in applications that require a sticky and relatively low-temperature modeling polymer, as well as one that is biodegradable.

The molecular weight of the obtained PHA was characterized by Gel Permeation chromatography (GPC), which demonstrated a retention time peak start at 14.07 min, peak maximum at 15.86 min, and peak end at 18.93 min (Appendix A). The mcl-PHA had average values, with a number average molecular weight (Mn) of 60,424, weight average molecular weight (Mw) of 114,093, Z-average (Mz) of 177,428, and viscosity average molar mass (Mv) of 105,309. The high polydispersity index value was because of the various monomer unit compositions and breakdown of the polymer in the sample preparation steps. The physical properties of the mcl-PHA produced by *P. resinovorans* differed from PHAs produced by other species [47]. Therefore, we hypothesize that the mcl-PHA has potential applications in the biomedical field or similar industries.

### 3.7. Mass Balance of Spent Coffee Grounds to Bio-Hydrogen and PHA

The mass balance was calculated on a carbohydrate basis to confirm the chemical oxygen demand (COD) reduction of SCGs (Figure 7). The 738 g COD of carbohydrates of the initial SCGs was reduced to 444 g COD in the residue. Then, hydrogen and PHA were produced. Hydrogen was 279 mL/g COD-consumed sugar, which showed higher productivity than the 115 mL–CH_4_/g COD using SCG hydrolysates reported in another anaerobic fermentation study [48]. In addition, the amount of PHA produced was 14 g/ kg SCG oil, which is a relatively low amount.

It is believed that this amount could be increased later through optimization of the culture process and production process. One of the important constituents in coffee is caffeine. Gokulakrishnan et al. reported that caffeine concentrations above 2.5 mg/mL can affect microbial growth [49]. Accordingly, it is necessary to reduce the amount of caffeine in SCGs. As shown by the mass balance, 80% of caffeine was removed in the residues.

However, after dark fermentation for hydrogen production, caffeine in effluent was 0.15 mg/g, which is about 6.5 times higher than the caffeine concentration of 23 mg/L at a sewage treatment plant in Seoul [50]. Caffeine did not cause a decrease in productivity during the hydrogen production process, but the amount of caffeine decomposition was very small. Therefore, it is necessary to establish a caffeine adsorption and removal facility in the post-treatment process when the operation is on an industrial scale.

## 4. Conclusions

This study produced eco–friendly energy and biodegradable plastics to induce the valorization of SCGs that are produced in large quantities every year. For bioplastic production, 97% oil was extracted from total oil in the SCGs. OESCGs were hydrolyzed and showed a sugar concentration of 32.26 g/L. OESCGH exhibited a low inhibitor concentration, and the post–treatment process could be omitted. Hydrogen production was 15% higher than that of SCGH. This shows that the process of extracting oil is more advantageous in the next utilization step. The extracted oil was used as a substrate for PHA production. *P. resinovorans*, which can use *n*–hexane remaining in the oil, produced 5.6 g/L of CDW and 29.7% of PHA content at 24 h in a fed–batch culture. In summary, it was shown that waste disposal costs can be reduced by performing both processes to increase the utilization of SCGs that were previously only hydrolyzed or had oil extracted. In addition, hydrogen and PHA were produced as useful resources. These results show that fossil fuel–energy and petroleum–based plastics can be replaced and enable the proposal of an environmental resource circulation model.

## Figures and Tables

**Figure 1 polymers-15-00681-f001:**
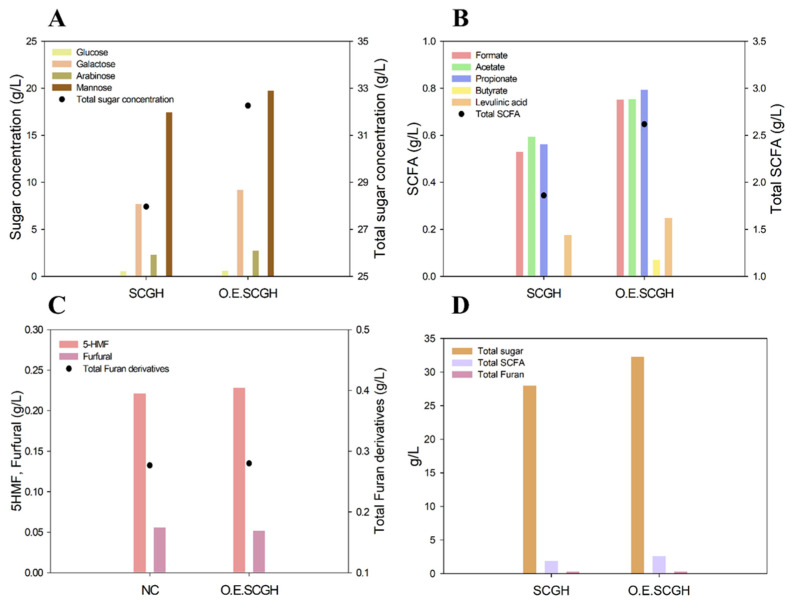
Comparison of spent coffee grounds hydrolysate (SCGH) and oil–extracted spent coffee grounds hydrolysate (OESCGH). (**A**) sugar composition; (**B**) SCFA composition; (**C**) furan derivatives composition; and (**D**) composition of sugar, SCFA and furan derivatives.

**Figure 2 polymers-15-00681-f002:**
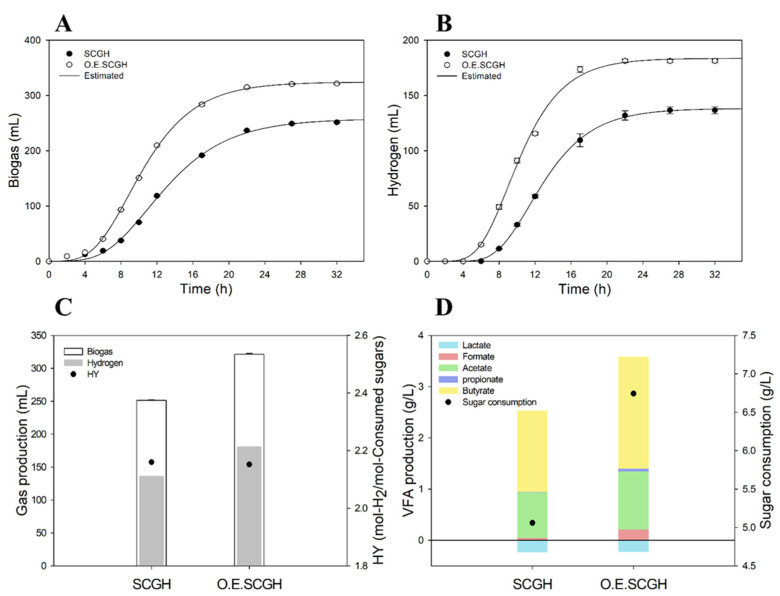
Comparison of dark fermentation productivities based on SCGH, OESCGH media (10 g/L of sugar concentration). (**A**) biogas profile; (**B**) hydrogen profile; (**C**) production amount of biogas and hydrogen, and hydrogen yield; and (**D**) VFA production and sugar consumption.

**Figure 3 polymers-15-00681-f003:**
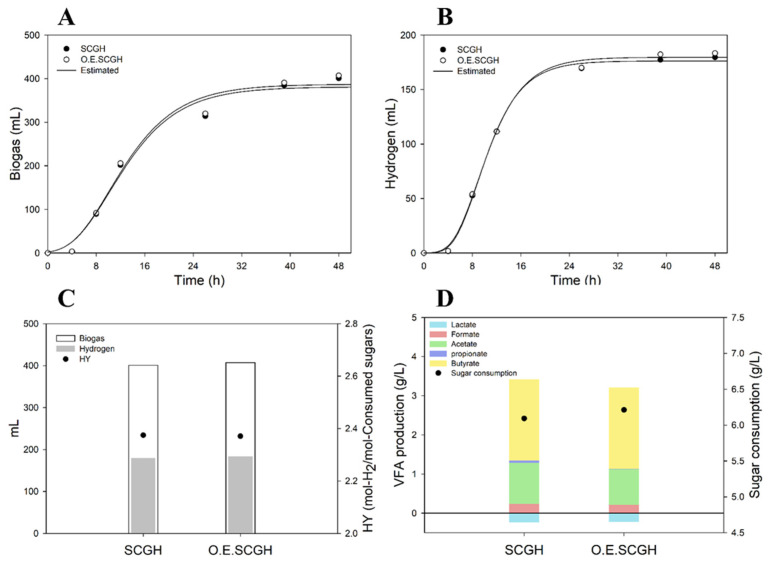
Comparison of dark fermentation productivities based on SCGH, OESCGH media (same volumes of hydrolysates in media [30% (*v*/*v*)]). (**A**) biogas profile; (**B**) hydrogen profile; (**C**) production amount of biogas and hydrogen, and hydrogen yield; and (**D**) VFA production and sugar consumption.

**Figure 4 polymers-15-00681-f004:**
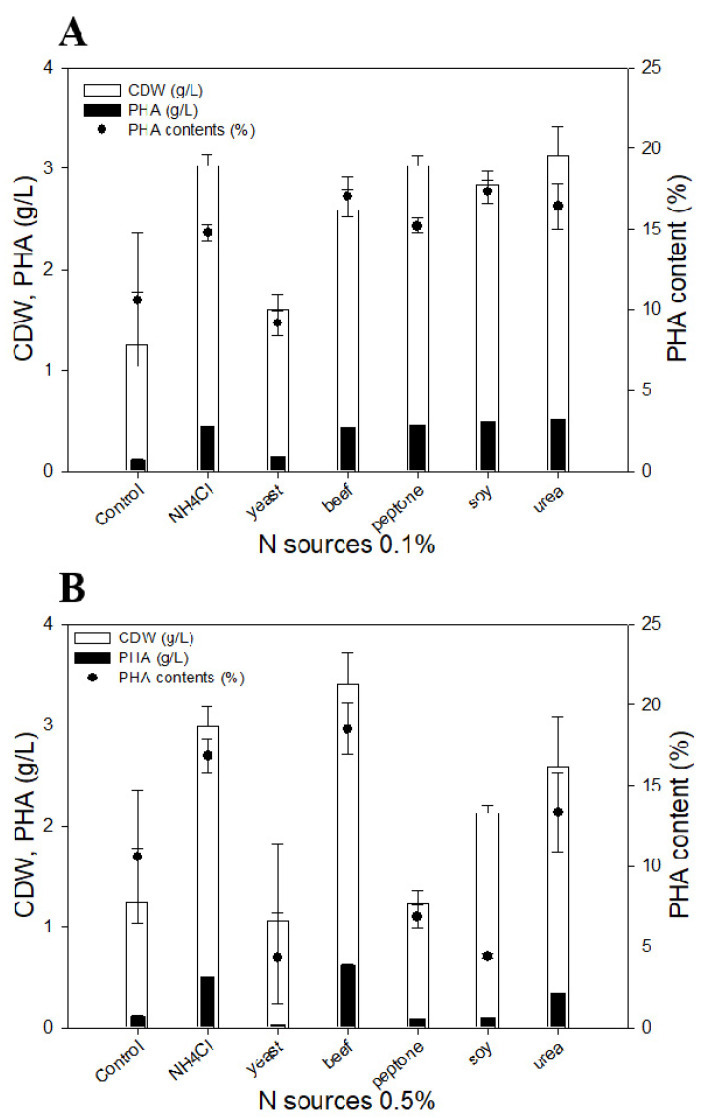
Nitrogen sources comparison for PHA production. Six different nitrogen sources were evaluated at 0.1% (**A**) and 0.5% (**B**) concentration; 2% coffee oil was used in the culture as a constant.

**Figure 5 polymers-15-00681-f005:**
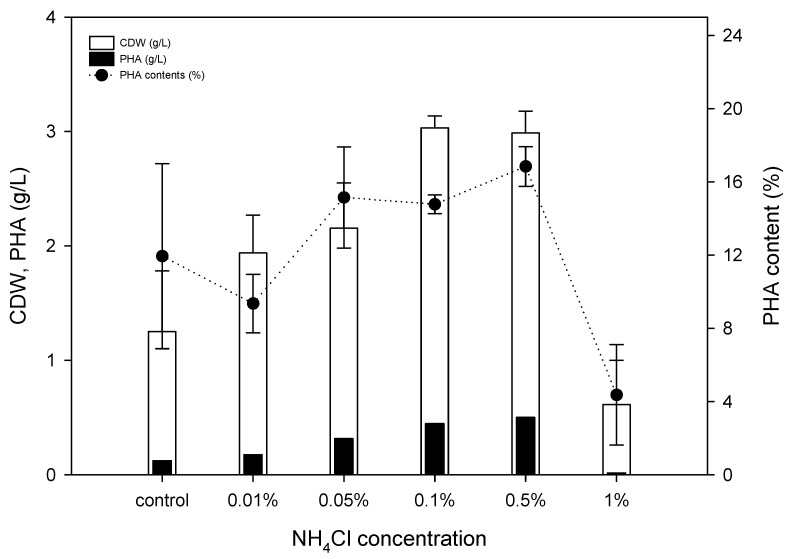
Effect of C/N ratio variance for PHA production. NH_4_Cl was used in the range of 0.01 to 1%, and 2% of coffee oil was used in the culture as a constant.

**Figure 6 polymers-15-00681-f006:**
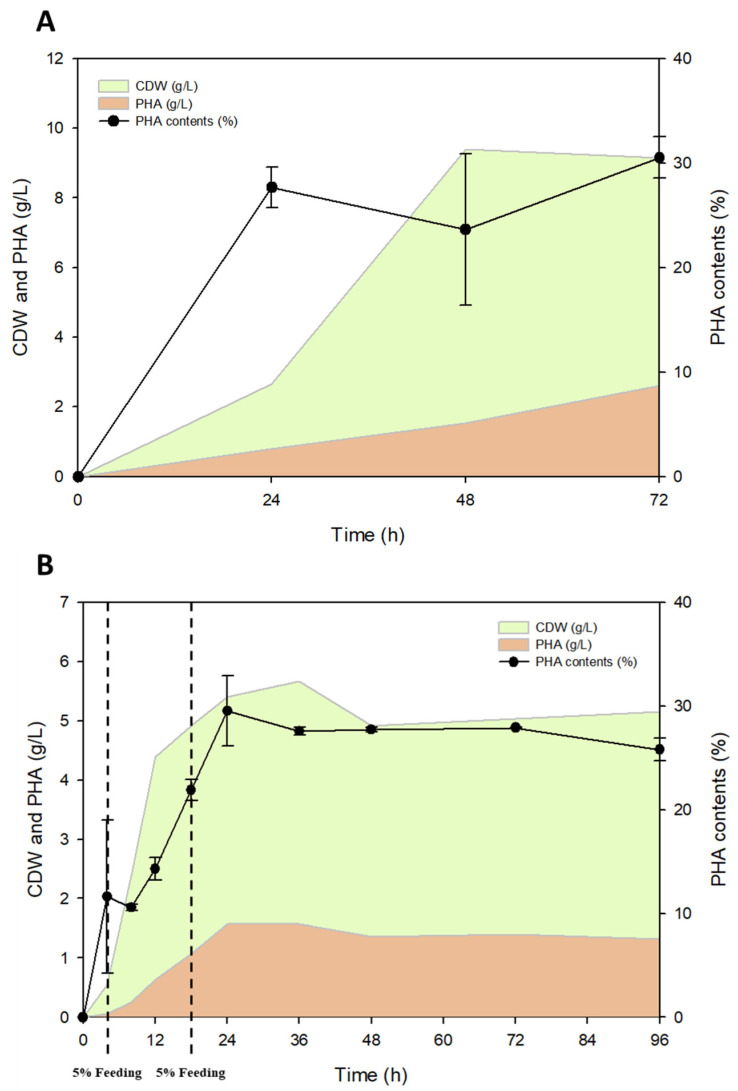
10 L scale jar fermentation for PHA production. (**A**) *P. resinovorans* was cultured with 2% of SCGs oil initially added as the sole carbon source on a 250 mL flask scale; and (**B**) *P. resinovorans* was cultured with 2% of SCGs oil initially added as the sole carbon source and 5% of SCGs oil fed sequentially after 4 and 18 h during the cultivation on a 10 L jar fermenter scale.

**Figure 7 polymers-15-00681-f007:**
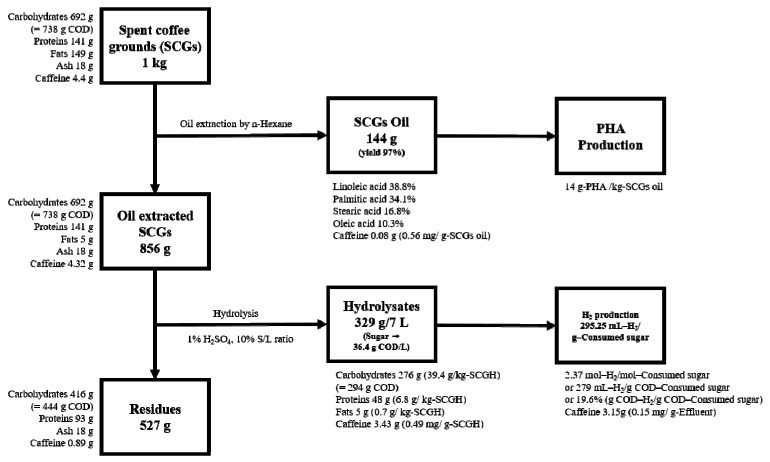
Mass balance of coffee grounds to bio-hydrogen and PHA.

**Table 1 polymers-15-00681-t001:** Conditions of spent coffee grounds oil and yield.

Extraction Conditions of SCGs Oil		Fractional Distillation of Extracted SCGs Oil		Oil Yield	
Hexane	200 mL	SCGs oil	4.03 g	Filter + SCGs	26.43 g
SCGs	25.00 g	Operation time	0.5 h	Weight of SCGs extract	3.62 g
Cellulose filter	5.05 g	Temperature	50 °C	Residual *n*–hexane	0.41 g
Operation time	1.5 h	Rotation	50 rpm	Oil yield	14.46%

**Table 2 polymers-15-00681-t002:** Chemical composition of spent coffee grounds.

Analysis Items	Results	Units
Carbohydrates	62.7	g/100 g
Proteins	12.8	g/100 g
Fats	13.5	g/100 g
Ash	1.6	g/100 g
Moisture	9.4	g/100 g

**Table 3 polymers-15-00681-t003:** Comparisons of hydrogen productivities using various biomass.

Biomass	Carbohydrates in the Hydrolysate	Organism	H_2_ Yield (mol-H_2_/mol-Consumed or Added Sugar)	References
Sugarcane bagasse	Glucose, Xylose, Arabinose	*Clostridium butyricum*	1.73	[14]
Rice straw	Xylose	*Clostridium butyricum* CGS5	0.76	[15]
Oil palm empty fruit bunch	Glucose, Xylose	*Enterobacter* sp. KBH6958	1.68	[16]
Corn stover	Glucose, Xylose, Arabinose	*Thermoanaerobacterium thermosaccharolyticum* W16	2.24	[17]
Oil-extracted spent coffee grounds	Glucose, Galactose, Mannose, Arabinose	*Clostridium butyricum*	2.37	This study

**Table 4 polymers-15-00681-t004:** Production of various PHA using SCGs oil.

Organism	Polymer	CDW (g/L)	PHA Contents (%)	PHA Concentration (g/L)	References
*Cupriavidus necator* H16	PHB	55.0	89.1	49.4	[41]
*Cupriavidus necator* DSM 428	PHB	16.7	78.4	13.1	[42]
*Ralstonia eutropha* Re2133	P(3HB-co-3HHx)	0.93	69	0.64	[43]
*Pseudomonas resinovorans*	Mcl-PHA	5.4	29.5	1.6	This study

## Data Availability

Not applicable.

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
