# Peer review of "Two-Stage Bio-Hydrogen and Polyhydroxyalkanoate Production: Upcycling of Spent Coffee Grounds"

_polymers, 2023, doi:10.3390/polym15030681_

Round 1
Reviewer 1 Report
The manuscript is entitled " Two-stage bio-hydrogen and polyhydroxyalkanoate production: Upcycling of spent coffee grounds" Authors have reported oil extraction from spent coffee grounds and used for polyhydroxyalkanoate and biohydrogen production. However, there are some points that need to be corrected. Therefore, recommended the publication of this paper after minor revision.
1. The material and method section should include a separate section that introduces the materials. Information such as the companies from which the materials used are purchased, their molecular weight, chemical properties, etc., should also be included.
2. Edit the H2SO4 spelling written on line 100. It should be written like this H2SO4.
3. Line 163,168: In chemical formula abbreviations, numbers are written under the letters. Please check the abbreviations of chemical formulas.
4. Line 260: Indicate the figure number in the sentence at the end of the sentence.
5. Line 346,347: Write the name of the bacterial species in italics.
6. Line 350: Check the beginning of the sentence.
Author Response
Reviewer 1
The manuscript is entitled " Two-stage bio-hydrogen and polyhydroxyalkanoate production: Upcycling of spent coffee grounds" Authors have reported oil extraction from spent coffee grounds and used for polyhydroxyalkanoate and biohydrogen production. However, there are some points that need to be corrected. Therefore, recommended the publication of this paper after minor revision.
R1Q1. The material and method section should include a separate section that introduces the materials. Information such as the companies from which the materials used are purchased, their molecular weight, chemical properties, etc., should also be included.
R1A1. We appreciate of the reviewer’s comments. The details of materials and chemicals which used in this study were included in the material and methods section.
R1Q2. Edit the H2SO4 spelling written on line 100. It should be written like this H2SO4.
R1A2. It was corrected according to reviewers commented.
R1Q3. Line 163,168: In chemical formula abbreviations, numbers are written under the letters. Please check the abbreviations of chemical formulas.
R1A3. It was corrected according to reviewers commented.
R1Q4. Line 260: Indicate the figure number in the sentence at the end of the sentence.
R1A4. It was corrected according to reviewers commented.
R1Q5. Line 346,347: Write the name of the bacterial species in italics.
R1A5. It was corrected according to reviewers commented.
R1Q6. Line 350: Check the beginning of the sentence.
R1A6. It was corrected according to reviewers commented.
Reviewer 2 Report
This is relatively a good paper. The authors presented a full work to show how they conducted oil extraction from SCGs and a fully utilization of SCGs via hydrolysis and fermentation. It is very useful to consider a recycling of SCGs. The paper can be considered for publication after very minor revision.
1. In the introduction, it seems insufficient to introduce other relavant work reported in the literature.
2. The authors presented many data regarding to oil extraction, hydrolysis or H2 production, but there is no mention or no comparison with others' work.
3. In Conclusion, how about the economic feasibility of this technical appproach?
Author Response
This is relatively a good paper. The authors presented a full work to show how they conducted oil extraction from SCGs and a fully utilization of SCGs via hydrolysis and fermentation. It is very useful to consider a recycling of SCGs. The paper can be considered for publication after very minor revision.
R2Q1. In the introduction, it seems insufficient to introduce other relevant work reported in the literature.
R2A1. We appreciate reviewer’s comments, and the explanation of relevant works were revised in the introduction.
R2Q2. The authors presented many data regarding to oil extraction, hydrolysis or H2 production, but there is no mention or no comparison with others' work.
R2A2. As reviewer mentioned, comparison data and its explanation were included.
R2Q3. In Conclusion, how about the economic feasibility of this technical appproach?
R2A3. As the reviewer suggestion, the economic analysis of hydrogen and PHA production using SCGs is very meaningful. However, in the case of mcl-PHA, downstream process has not been established and there are no commercialized products yet, so it is difficult to estimate the production cost. Therefore, this study focused on hydrogen and PHA production and possibility using SCGs. Unfortunately, it would be difficult to contain what the reviewer mentioned in this paper, but we expect that economic analysis is also possible as downstream process are established.